# Extractivism and Unjust Food Insecurity for Peru's Loreto Indigenous Communities

Caterina Rondoni

Environmental Sustainability and Wellbeing, Department of Humanities, University of Ferrara, 44121 Ferrara, Italy; caterina.rondoni@unife.it

**Abstract: Background**. Many scholars have examined Indigenous food security and sovereignty yet the topic still represents a small share of environmental justice scholarship. Therefore, we completed a case study of the environmental justice challenges concerning food security faced by the Indigenous communities of Peru's Loreto region. **Methods**. During 2019, we conducted fieldwork in 64 Indigenous communities of Kukama Kukamiria and Urarina in the Amazon rainforests of Loreto, Peru. Based on a semi-stratified sample and snowball sampling method, we combined participant observation with 139 interviews focused on feeding habits, production and availability, access, utilization, food stability, and perception of food insecurity with the Food Insecurity Experience Scale (FIES) method. **Results**. Analyzing these themes led to worrisome assessments of the food insecurity and institutional limits of Indigenous communities. Because of their geographic location, these communities experience a degraded and unhealthy environment with water and food contaminated by hydrocarbon extraction activity. Furthermore, Peru's policy of food and nutrition security has public management deficiencies especially in the Loreto region. Thus, many of the efforts adopted remain ineffective. **Conclusion**. Indigenous communities that live following ancestral culture often lack resources to change their diets. Thus, they frequently suffer the most following the contamination of an environment with which they experience an interdependent relation.

**Keywords:** food security; food insecurity; environmental justice; Indigenous communities; oil pollution; Latin America

## 1. Introduction

Food is an almost infinite universe of connections, references, concatenations, and influences between entirely different areas, and access to it has always conditioned individual and social behaviors [1]. Food is also one of the fundamental human rights, recognized in all the main conventions and treaties concerning people's rights since 1948. Inside a community, it is one of the strongest elements of collective identity. Indeed, food represents an indicator of the social group it belongs to, especially when we speak about Indigenous peoples [2]. However, the question is: what happens to this fundamental right in the event of serious pollution?

Research from Lancet Commission on Pollution and Health found pollution, especially in low- and middle-income countries, to be the largest environmental cause of death and disability in the world. In 2015, diseases caused by pollution were responsible for 9 million premature deaths; that is 16% of all global premature deaths. According to the Lancet Commission on Pollution and Health [3], exposure to contaminated air, water, and soil kills more people than a high-sodium diet, obesity, alcohol, road accidents, or child and maternal malnutrition. Furthermore, the Commission added that pollution disproportionately impacts low- and middle-income countries: for example, oil pollution disparities are seen in Nigeria's Niger delta region, Ecuador's Oriente basin, and Peru's Loreto region [4]. Therefore, these populations, already affected by economic discrimination and lack of services, suffer unequal exposure to contamination and environmental risks.

We are thus faced with what is called distributive injustice, one of the three pillars employed commonly by EJ scholars. This injustice, experienced mainly by Indigenous people, is related to the inequitable distribution of environmental goods (such as clean water, air, food) and environmental risks (such as exposure to pollution and other hazards). The two other dimensions of EJ are procedural injustice and recognitional injustice. The former is correlated with procedural barriers that Indigenous communities have to face when it comes to the possibility of their being involved in planning, managing, and governing their ancestral lands, waters, and seas. Recognitional injustice refers to the misrecognition and general disrespect for Indigenous groups' cultural identities, values, and knowledge systems but can also include the misrecognition of fundamental rights related to the culture, such as the right to land and water. As Parsons stated in his work [5], this "lack of knowledge" of the makes it easier for governments and companies to justify locating environmental hazards such as toxic waste disposal sites, nuclear power plants, oil and gas refineries, and mining operations on Indigenous lands and close to Indigenous settlements or on sites of cultural significance.

This article seeks to improve our understanding of the relationship between extractivism and food security as a challenge for the fulfillment of environmental justice (EJ) via a case-study of Peru's Loreto Region. It will do so by using the Food Insecurity Experience Scale method (FIES), a method developed by Food and Agriculture Organization of the United Nations (FAO) [6]. The aforementioned method produces a measure of the severity of food insecurity experienced by individuals or households based on direct interviews. Many studies use FIES as a methodology to measure food insecurity, but this is one of the first to apply this approach to Latin American Indigenous communities. Indeed, after conducting a literature review using all databases available within the Web of Science (WoS), we found that literature on food security (34,735 articles in WoS) and environmental justice (5955 articles in WoS) rarely overlap. Using the Topic Search (TS) terms "Food Security" AND "Environmental Justice" (query: (TS = ("Food Security")) AND TS = ("Environmental Justice")), the research yielded 31 results. Thus, less than 0.1 percent of the literature inventoried by the WoS explicitly integrates food security with EJ. With the additional search term of "Latin America" added to our query (TS = ("Food Security") AND TS = ("Environmental Justice") AND TS = ("Latin America"), the search yielded one result. Finally, adding the search term "Indigenous Peoples" to our query (TS = ("Food Security") AND TS = ("Environmental Justice") AND TS = ("Indigenous Peoples"), the search yielded again only one result while using the whole query (TS = ("Food Security") AND TS = ("Environmental Justice") AND TS = ("Latin America") AND TS = ("Indigenous Peoples"), we obtained no results. Comparing our results using all four search terms (EJ, Food Security, Latin America, Indigenous Population) to the total hits when searching for "EJ" and "Food Security" reveals that no food security literature integrated with EJ and inventoried in WoS has a particular focus on Latin America and the Indigenous populations.

Although the WoS does not provide an exhaustive literature review on a topic (other articles on this topic can be found on Google Scholar, for example), the results are scholarly and reproducible. Thus, this paper contributes to an underexplored subtopic of food security.

### 1.1. Food Security and Right to Food in Peru

Food security is a concept that is linked to many issues: agricultural production, trade and investment, quality of food and water, climate change, food waste and loss, access to land, and management of natural resources [7]. According to FAO, "food security exists when all people, at all times, have physical, social, and economic access to sufficient, safe, and nutritious food that meets their food preferences and dietary needs for an active and healthy life" [8]. Therefore, food security is much more than consuming nutritious and healthy food. It involves a close interrelation between security and safety (a sum of quantitative and qualitative dimensions). Thus, while the pillars of availability, access,

and stability are associated with the scope of security, the pillar of utilization is closely linked to the nutritional dimension, related to the concept of food safety and of a more qualitative nature [9]. Therefore, the discussion around the right to food and adequate nutrition should not be approached as a one-dimensional issue managed by a single state sector. On the contrary, due to its complexity, it should be assumed through joint work between all the actors involved.

But food security is just one of the aspects of the broader right to food, also recognized on paper by the Peruvian state. However, the Peruvian constitution does not explicitly include this right but implies it in article 2.1: the right to life and moral, mental, and physical integrity [10]. Likewise, the 1993 Peruvian constitution establishes a chapter on fundamental rights without including economic, social, and cultural rights [11]. Despite this uncertain national situation, over the last fifteen years there has been an abundant drafting of legal norms, administrative provisions, plans, programs, and ministerial policies that deal with the right to food and, more precisely, to food security [12]. Among these, the most important are: Plan Nacional de Derechos Humanos 2006–2010 (Decreto Supremo N° 017-2005-JUS); Ley de Inocuidad de los Alimentos (Decreto Legislativo N° 1062-2008) and Estrategia Nacional de Seguridad Alimentaria y Nutricional del periodo 2013–2021 (Decreto Supremo N° 021-2013-MINAGRI).

However, these policies have not established legal norms recognizing this right in Peru or ensuring food security [12]. What is lacking is a specific law [13] and a regulation that connects and organizes the different aspects of the right to food and identifies specific competencies for the various actors who deal with this issue. Despite many lower-level regulations and official programs oriented to the enforcement of food safety, these mechanisms are not enough to achieve the organization of an articulated space capable of solving a problem as complex as that of food insecurity.

In the Loreto region, where we decided to develop the case study, the protection of the right to food is encountering some difficulties in achieving a well-defined workspace. With the publication of the "Regional Strategy for Food and Nutrition Security of the Loreto Region" (Regional Ordinance No. 017-2012-GRL-CR), 2012 was a good year for food security in the region: not only did they begin to think about the problem of access to food and its complexity, but an attempt was made to involve the various actors responsible for the issue. Unfortunately, the project did not continue. Despite the efforts made in recent years by the various institutions involved to achieve the objective of food security, only recently, with the "Regional Strategy for Comprehensive Early Childhood Care" (Regional Ordinance No. 005-2019-GRL-CR), did they begin to think again about a project undertaken jointly by the different institutions.

### 1.2. Regional Context

Located in the extreme northeast of Peru, the region of Loreto (Figure 1) is divided into 8 provinces and 53 districts where 883,510 inhabitants live, that is 3% of the total national population registered. With an area of 368,852 km$^2$, Loreto represents 28.7% of the national territory; it is Peru's largest region and it is bigger than Ecuador and Germany. Loreto belongs to the so-called "Llano Amazónico" and 95% of its territory is cover by primary rain forest [14].

The highest percentage of Peru's Indigenous population (43.2%) lives in the Loreto region spread out over 28 native communities according to the Instituto Nacional de Estadística e Informática (INEI) [15]. Due to the high degree of human and natural diversity and its geographical location, this region presents a rather complex social and economic structure.

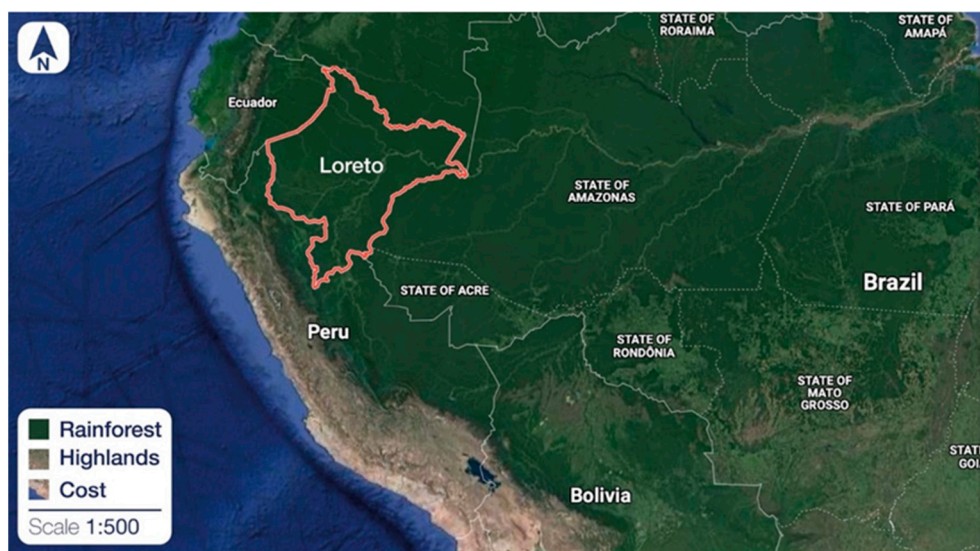

**Figure 1.** Loreto region (source: Image @2021 NASA, TerraMetrics, Maps Data @2021 Google).

Access to education in Loreto is deficient, resulting in an enormous gap in access to services between urban and rural inhabitants and between men and women. At the urban level, 3.99% of males lack education; at the rural level this number grows up to 9.03%. Education among women in Loreto is even lower: 4.9% of urban women lack education while 14.4% of rural females are uneducated (Figure 2 and Supplementary Materials, Section SA, Table S4).

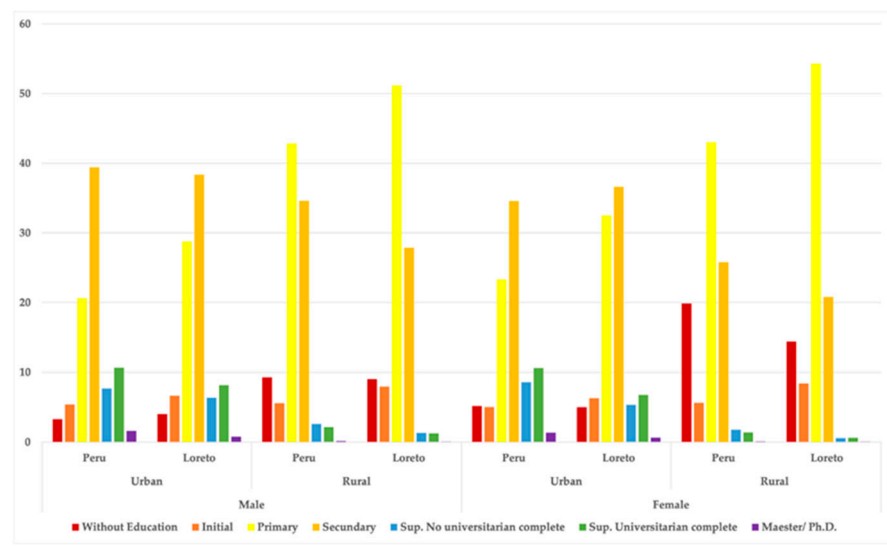

**Figure 2.** Census 2017 population (%) aged three and over by sex as well as urban and rural area according to educational level attained, national and regional comparison (source: own elaboration from the data of the INEI, Loreto. Compendio Estadístico 2017).

Female healthcare and health disparities are also evident. There is a 79.7% level of prenatal and child delivery care in Loreto compared to 96.9% nationally. Likewise, teenage pregnancy rates reach 20.4% in Loreto compared to only 14.6% nationally (Supplementary Materials, Section SA, Table S6 and Figure S1).

The Intituto National de Estadistica e Informatica (INEI) reports that the agricultural and fishing sector continues to constitute most of the region's employment (32.95% of the census population). While only 0.18% of the population carries out activities related to the

exploitation of mines and quarries, this sector represents the largest source of wealth in the region (Supplementary Materials, Section SB, Table S9, and Figure S2) [16].

### 1.3. Oil Development in Loreto

The Peruvian Amazon rainforest began to attract the interest of the state around the 1970s after the discovery of several oil fields. Today, more than 60% of the national oil production and almost 100% of natural gas comes from the jungle. In particular, the Loreto region hosts Peru's largest proven onshore oil reserves predominantly within oil Blocks 8 and 192 (formerly Block 1AB) [17]. In 1972, the State commissioned PETROPERÚ S.A. [18], a subsidiary of the Ministry of Energy and Mines, to manage the construction of the pipeline that today transports oil for 845 km. Over the years, the role of the Amazon in energy production grew to include 74% of the national territory granted for energy extraction in 2014 [19]. At the same time, the social and environmental problems associated with this activity have grown, especially because of the pipelines, their age, and poor maintenance, as shown in Schemes 1 and 2.

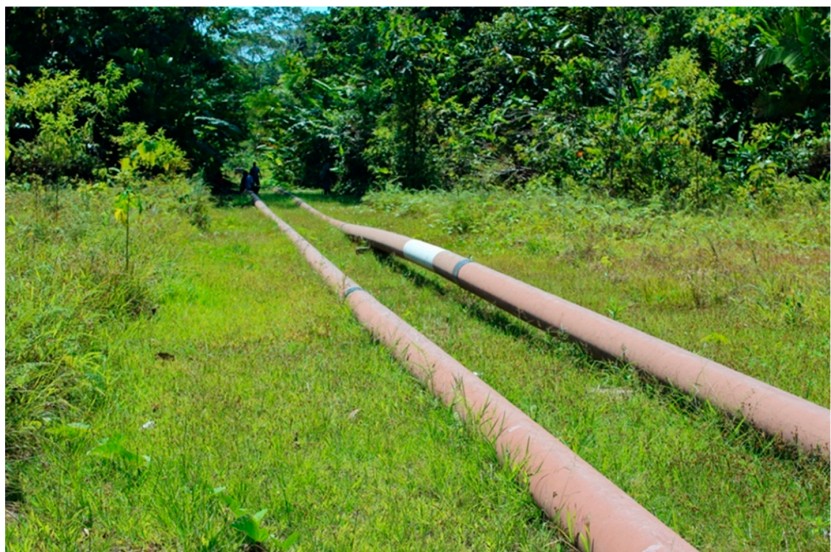

**Scheme 1.** View of the North Peruvian pipeline near La Petrolera village (photo by author took in August 2019).

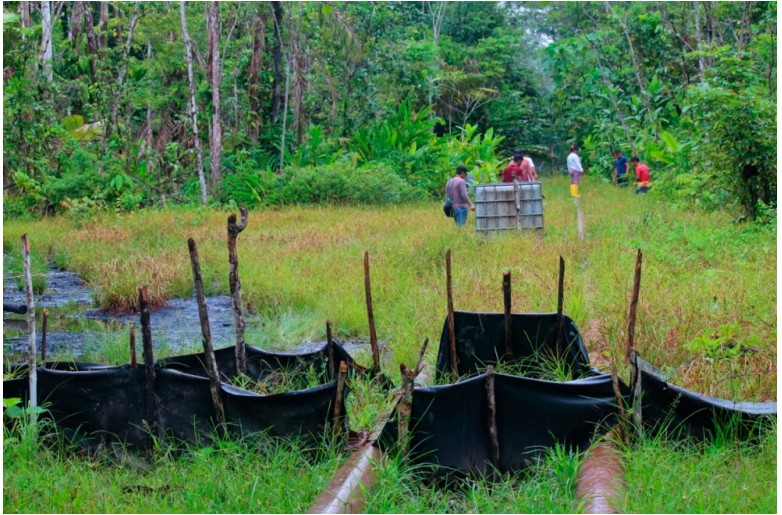

**Scheme 2.** View of the North Peruvian pipeline oil spill location near La Petrolera village (photo by author took in August 2019).

Several news sources reported severe oil spills in the Amazon rain forest (Figure 3). In 2016, for example, 3000 barrels of oil (about 477,000 L) contaminated at least 30 km of the Chiriaco River. The spill also spread into the Marañón River, one of the main tributaries that feed the Amazon drainage area [20]. A 2020 published report entitled "The shadow of oil" (La sombra del petróleo) stated that 474 oil spills occurred in the Peruvian Amazon between 2000 and 2019. The report also determined that 65% of these spills were caused by the corrosion of oil pipelines and operational failures [21].

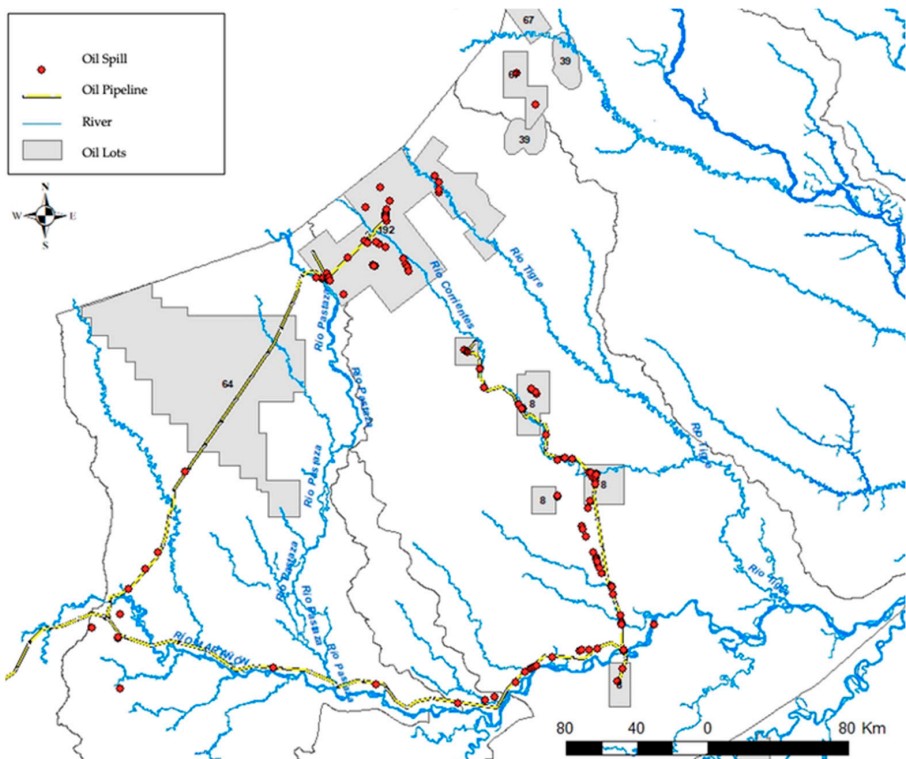

**Figure 3.** Map of oil spills that occurred in the North Peruvian Oil Pipeline in the Loreto Region from 2011 to 2019 (source: Organismo de Evaluación y Fiscalización Ambiental—OEFA).

Thus, oil pollution externalities in northern Peruvian Amazon concessions have compromised the integrity of the environment throughout the territory. Oil infrastructure also leads to other negative impacts. For example, researchers documented deforestation from the construction of access roads, drilling platforms, and pipes. Oil spills also result from produced water discharges [22]. More negative consequences result from ineffective and incomplete environmental remediation by extractive companies which often try to hide the damage or disregard their responsibilities.

Together with hydrocarbon extraction operations, oil companies discharged three million barrels of produced wastewater into water sources used by regional populations [23,24]. These studies also documented how these waters are loaded with high levels of chlorides, dissolved hydrocarbons, organic acids, phenols, and metals such as cadmium, chromium, lead, barium, and carcinogenic, or neurotoxic substances coming from toxic waste. Another investigation documented how produced wastewater discharges stopped only in 2009, when scientific evidence demonstrated the risks of this practice for both the environment and human health [25]. Consequently, contamination from decades of oil extraction impacted some of the food supply not only for Indigenous communities but for the entire Loreto region.

A few studies examined the impact of oil extraction on the Loreto region. However, these are scarce. First, an interagency commission in 2012 analyzed, designed, and proposed remedies for the social and environmental conditions of the Loreto communities located

along the Pastaza, Tigre, Corrientes, and Marañón rivers [26]. The results showed that people's health and lives were at risk due to the contamination generated by Pluspetrol's oil activities during the 13 and 17 years of concession of lots 192 and 8, respectively. For this reason, the government was forced to declare a state of environmental emergency in the basins of the Pastaza, Corrientes, and Tigre rivers in 2013 and in the basin of the Marañón River in 2014 (5 May 2014, RM N° 136-2014-MINAM) [27]. In that same year, a state of sanitary emergency was declared for all four hydrographic basins [28,29].

Moreover, a large research team led by researchers at the Universitat Oberta de Catalunya in Barcelona, Spain, [30] found high levels of lead in the bodies of Indigenous people living in the Peruvian forest near areas where oil extraction takes place. The authors hypothesized that the metal reached Indigenous people through their diet in areas with the most serious environmental pollution. Indeed, the population hunts and fishes for food, and previous studies showed that lead is present in animals in this region [31]. On the other hand, in places with lower levels of environmental pollution, the most likely route is occupational exposure such as coming into direct contact with oil due to participating in clean-up tasks after oil spills. According to the World Health Organization, exposure to lead could cause alterations in the nervous, hematological, gastrointestinal, cardiovascular, and renal systems and, therefore, puts people's health and lives at risk [32].

The Barcelona University team has studied a population that live in the basins of the Corrientes, Pastaza, and Tigre rivers, all major tributaries of the Marañón River which forms the headwaters of the Amazon River. The highest levels of lead in the blood were found among participants coming from the Corrientes River basin which accounts for most of the oil extraction activity in the region. The study also found higher levels of lead in the blood of people living less than an hour's walk from an oil facility. The values observed in this study are twice as high as the values reported for children in Europe between 1999 and 2007, at a time when leaded petrol was still used in Europe (which in some countries continued until 2005). The results showed also higher levels of lead among males, caused by their higher involvement in activities that expose them to lead such as cleaning up spills.

### 1.4. ACODECOSPAT and Its Basis

In 2000, around 11 communities in the lower part of the Marañón river (see Figure 4) formed the Indigenous advocacy organization Asociación Cocama de Desarrollo y Conservación San Pablo de Tipishca (ACODECOSPAT). Today, the number of communities who are part of the organization has risen to 64, composed mainly by two Indigenous populations: Kukama-Kukamiria and Urarina. In order to understand why oil pollution is a major threat for these communities it is important to underline the relation that these Indigenous populations have developed with the rivers. This is a perspective that, according to Parsons' work, scholars often seems to misrecognise. Instead, researchers apply their cultural perspectives while ignoring or dismissing ancestral ones [5].

Most of Kukama-Kukamirias live in their own communities in hamlets or on the banks of rivers and lakes, working in subsistence agriculture and fishing. In addition, due to their long interrelation with a floodplain ecosystem and their great adaptation to it, the Kukama-Kukamirias have developed different fishing tools and techniques, which today are valuable cultural practices inherited from their ancestors [33]. All productive activities such as fishing, agriculture, hunting, and gathering, and transportation of products and people depend on the fluctuations of the river that regulates the annual cycles of plant and animal life, and the possibilities of subsistence. Therefore, it is not surprising that the most significant representations of their material and spiritual world are linked to the river [34]. All the elements of Kukamas culture show the centrality of fish and rivers for this people [33]. Indeed, for the Kukamas, in addition to the worlds of heaven and earth, there is the world of water; Kukamas consider the entire river to be sacred and inhabited by large and small populations of legendary creatures as well as by relatives who disappeared from the earthly world [35].

Like the Kukamas, the Urarinas have also developed a deep knowledge of floodplain ecosystems. This knowledge has forged a series of skills and abilities that allow an efficient use of resources. But they have also generated a structure of values, beliefs, and cultural practices that has strengthened the bond of the Urarinas with their environment, making them its best keepers. The presence of spiritual beings who inhabit the invisible plane of reality, recognized by most of the Amazonian Indigenous peoples as the owners, mothers, or grandparents of plants and animals, determines a social relationship that goes beyond the simple use of a natural resource, being one of the factors that favors conservation in Indigenous territories. The belief in beings like the baainu [36], which inhabits the wetlands [37] that surround the Urarina communities, is a form of social control that favors the conservation of traditional ecosystems such as the jiiri and the alaka. The intergenerational transmission of knowledge ensures the survival of Urarina communities but it is also the best tool to understand changes in the environment caused by climate variability or human pressure [38].

The aforementioned relation between rivers and oil contamination has made extraction activities by the oil companies Pluspetrol and Petroperu a primary concern of ACODECOSPAT. In recent years, the organization highlighted the mismanagement of oil pipelines and infrastructures and the importance of protecting forests, rivers, land, plants, animals, and spiritual beings. The organization then focused on articulating and protecting Indigenous rights and community benefits. Currently, 64 native Kukama and Urarana communities form a federation protecting villages on the banks of the Marañón, Nahuapa, Ucayali, Chambira, Patoyacu, and Amazonas rivers [39].

Below we will proceed to a more in-depth study of the food situation of the 64 communities that are part of the federation.

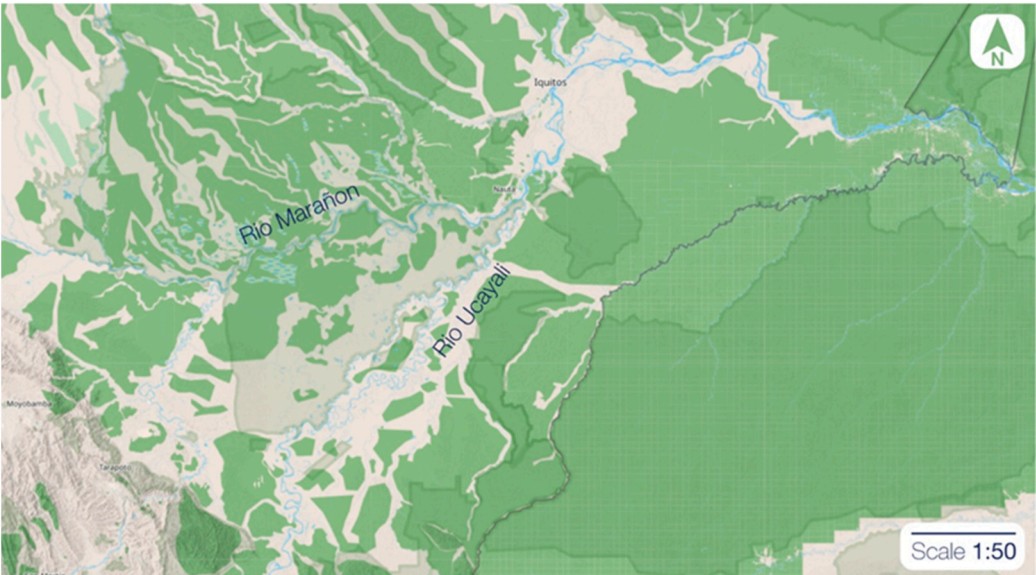

**Figure 4.** Marañón and Ucayali rivers (source: OpenStreetMap contributors).

## 2. Materials and Methods

Similar to Gonzalez's previous study on the political ecology of Peru's hydrocarbon development [4,16,40], this author relied on a qualitative case study in order to collect primary data through a multi-method qualitative approach incorporating semi-structured interviews and participant observation. Following Gonzalez's work but with an additional focus on food insecurity, a seven months' fieldwork was conducted in the Loreto Region of Peru.

Direct interviews were the main tool chosen to collect data related to eating habits, food availability, access to food, and other information related to the community structure

in general. The decision to open a direct dialogue and not administer a questionnaire was made with the goal of collecting not only specific information but also a broader vision on the issue of food. Furthermore, by doing so, it was possible to overcome the difficulties encountered when administering a written questionnaire to Indigenous community members of any age. We chose to follow the same structure for each interview, providing a set of closed and open questions to be administered following a logical scheme (Supplementary Materials, Section SC). The interviews carried out lasted between 30 and 40 min.

Interview questions were based upon a pre-field interview guide tested through several pilot interviews and refined throughout fieldwork [41] and they were carried out in Spanish, a language spoken by all the sample components. The interviews were subsequently systematized using Microsoft Excel, following a query system divided thematically and sub-thematically similar to the structure of the questionnaire (Supplementary Materials, Section SC). Finally, primary data collection was supported by secondary-source-based analysis of Peru's political environment. This included analysis of Peru, regional and international jurisprudence, and data statistics, all specific to the production of food and hydrocarbon.

Concerning the sample choice, we focused our work on the ACODECOSPAT Indigenous federation, made up of 64 communities distributed in the 4 Loreto districts of Maquia, Nauta, Punchana, and Urarinas along the rivers Amazon, Chambira, Chiriyacu, Marañón, Patoyacu and Ucayali. The decision to choose this group of communities as a sample was made in accordance with the interest shown by ACODECOSPAT in issues such as the protection of the environment and human rights, including the right to adequate food. Initial contact was made thanks to the Centro Amazónico de Antropología y Aplicación Práctica (CAAAP), a Peruvian NGO founded in 1974 and working since its inception with Peruvian Indigenous communities on equity, interculturality, human rights, and the environment [42].

Visiting all the federation's communities was impossible with limited time and resources so a representative sample of the communities was selected according to their internal composition and geographic distribution. The semi-stratified (Table 1) sample was obtained dividing the communities into 5 strata based on the following variable: geographic location of the community, belonging to a certain Indigenous group, gender, role covered within the community, and job performed. Because of the difficulty of the interaction within Indigenous communities and the geographical characteristic of the region, we adopted a snowball sampling method [43]. From each community we started from key informants of the community, remaining faithful to the constraints dictated by the subdivision by strata.

**Table 1.** Sample Summary.

| Analysis Unit | Individual | | |
|---|---|---|---|
| Sampling | 39 out of 64 communities | | |
| Informant | 139 individuals<br>- Female: 60<br>- Male: 79 | | |
| Key Informants | - Apu<br>- Environmental monitor<br>- Indigenous mother | | |
| Other informants | - Health promoter<br>- Wise | - Farmer<br>- Fishermen<br>- Hunter | |

The result is a sample of 139 people (60 women and 79 men) belonging to 39 communities of the federation, interviewed from April to October 2019. Key informants, i.e., collaborators who can provide primary data because they have a good amount of knowledge about their community, were chosen for each. In our case these figures were: the Apu, legal representative of the community; the environmental monitor, an expert with knowledge of their territory; the Indigenous mother, representative of women; and, when present, the health promoter and the wise man of the community. Starting from the key informants, an attempt was made to choose male and female interlocutors inside the strata as well as representatives of the different categories of fishermen, hunters, and farmers.

The fieldwork was carried out over a period of 7 months (30 April 2019–23 October 2019) during which 7 field trips were completed, with an average duration of 4 days each, visiting a total of 8 communities. During these outings, different useful spaces were set up for carrying out the research: direct interviews, visits to the field, trips to the river to carry out fishing activities with the community members, and visits to the kitchens to assist in the preparation of food. Some of these field trips coincided with ACODECOSPAT federation gatherings. These events were the opportunity to get in contact with people from very distant communities selected in the sample with whom, otherwise, it would have been impossible to get in touch. Indeed, due to the lack of means and the difficult geographical access, some representatives of Indigenous communities do not have the possibility to leave their communities often and share the needs of the other members.

This author is aware that fieldwork was undertaken with only two Indigenous ethnic populations in Loreto, both belongings to the same Indigenous federation. Other communities' experiences, geographical location, and culture are different, so this one study does not necessarily represent the food situations of all Indigenous communities in Loreto.

Moreover, due to the geographic and social peculiarities of the Loreto region, there is a lack of homogeneous data and statistics on food. Cultural dimensions also posed challenges for the research. Therefore, the present study followed previous food security studies in Latin America [44–46] by utilizing the Food Insecurity Scale (FIES). FIES is a method developed by FAO that produces a measurement of the severity of food insecurity experienced by individuals or households based on direct interviews. The tool is designed for cross-cultural equivalence and validity in developing and developed countries to produce comparable indicators of the prevalence of food insecurity in a given population.

Through data collected during direct interviews regarding people's limitations to obtain adequate food, this indicator makes it possible to estimate what portion of the analyzed population faces difficulties in accessing food and at what level of severity. This measure was created in order to have an indicator capable of overcoming the concept of hunger associated with under-nutrition and over-nutrition and detect the experience of hunger from the beginning. Indeed, before a person is classified as being undernourished, they start to worry about having enough food, they adopt dietary changes to make limited food resources last, and finally they decrease food consumption in the household.

Measures such as undernourishment and anthropometric measures (child weight-for-age and height-for-age) provide invaluable information regarding the nutritional status of individuals but are costly and require a relatively sophisticated level of expertise to collect and analyze the data. Instead, experience-based food insecurity scales like the FIES represent a simple, time-efficient, and less expensive method to measure food insecurity based on data collected at a household or at an individual level.

Unlike aggregate measures such as the FAO Prevalence of Undernourishment or the Global Hunger Index, the FIES measurements of the severity in food insecurity can be used in surveys that allow disaggregation at sub-national levels and across different populations, making it possible to identify more accurately who the people with food insecurity are and their geographic distribution, thus identifying vulnerable populations before malnutrition becomes manifest. Finally, the ease of application, analysis, and interpretation facilitates communication of results to decision makers, leaders of civil society, and the general public [6].

The items that compose the FIES module directly ask people about the quality and quantity of the food they eat and why they eat that way, be it limited money or other resources. By asking this series of eight questions, it is possible to classify respondents at different levels of severity of food insecurity (Figure 5).

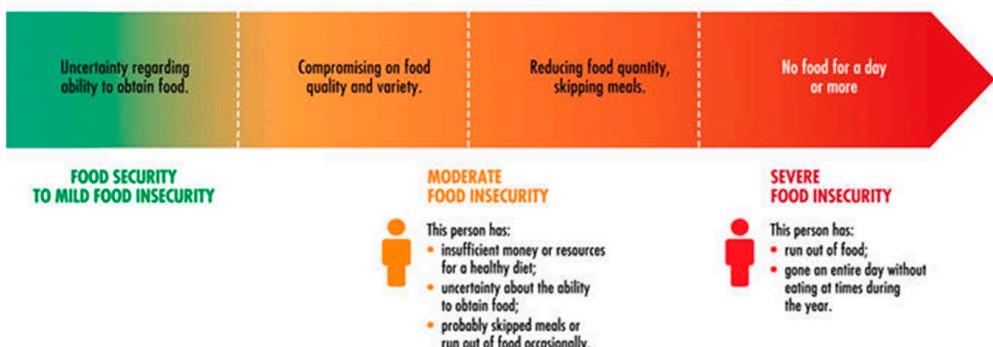

**Figure 5.** Food insecurity based on the FIES (source: FAO, The Food Insecurity Experience Scale Development of a Global Standard for Monitoring Hunger Worldwide, Rome, 2013).

Figure 5 points to one of the unique contributions of experience-based food insecurity scales. In addition to contemplating aspects related to deprivations in diet quality and quantity, they also capture an aspect of the experience of hunger and food insecurity that others do not, i.e., psychosocial aspects associated with anxiety or uncertainty regarding the ability to procure enough food. The ordinal scale includes three categories: (1) mild to moderate food insecurity, which is typically characterized by uncertainty and anxiety regarding access to food and changes in diet as the situation worsens, such as a less balanced and more monotonous diet; (2) moderate to severe food insecurity, where the amount of food consumed decreases (e.g., portion sizes are reduced or meals are skipped); and (3) severe food insecurity, characterized by feeling hungry or not eating for an entire day due to lack of money or other resources.

As shown in Figure 6, food insecurity can affect health and well-being in many ways, with potentially negative consequences for mental and social health in addition to physical well-being, even in the absence of measurable negative effects on nutritional status.

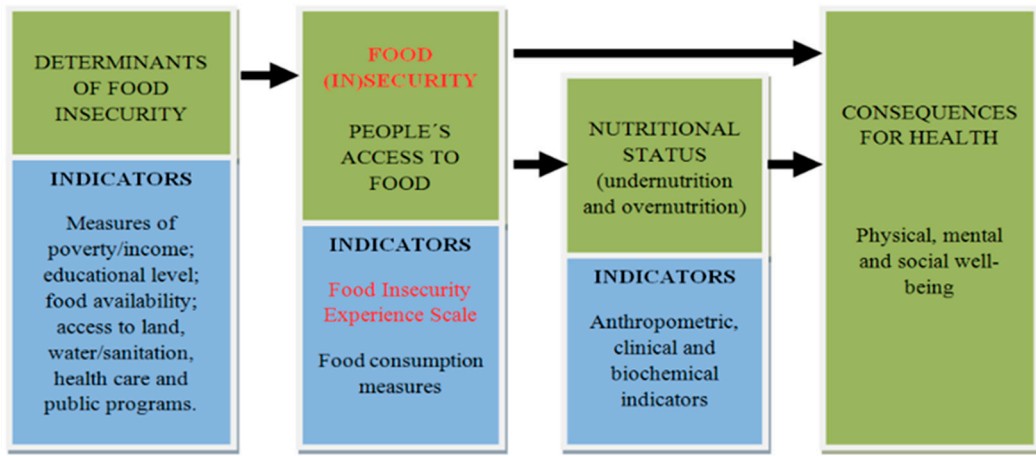

**Figure 6.** Determinants and consequences of food insecurity at an individual level (source: FAO, The Food Insecurity Experience Scale Development of a Global Standard for Monitoring Hunger Worldwide, Rome, 2013).

Fundamental in the choice of using the FIES method was the evident lack of data collected by national agencies on food insecurity in Loreto. This shortcoming in data collection is even more evident when we take Indigenous peoples into consideration. We

often come across cases of Indigenous peoples not registered at the civil registry office who, practically speaking, do not exist for the state. Considering the complex and multicultural context in which the study took place, FIES was found to be the most effective—if not the only—tool that could be used. Since this method is based on the perception of hunger, it was easy to use it in a complex context such that of the ACODECOSPAT federation of Indigenous communities. Thanks to the simple questions part of FIES, it was not necessary to explain to the interviewees the complex meaning of "food security", thus avoiding direct, although unintentional, answers and comments. All interviewees, regardless of their role in the community and the degree of education they possessed, were able to answer the questions without difficulty, guaranteeing comparable data for the research.

## 3. Results

### 3.1. Feeding Habits

The diet constitutes the alimentary repertoire that a society selects between the different possibilities that are presented to it. Although it is conditioned by the availability of food in a given place, either because it is produced there or because it is easily obtained and transferred from other areas, the diet is also influenced by other factors such as the history of the group and the relationship with other groups, changes in certain elements of culture, and changes in the natural environment. A significant percentage of the **Kukama-Kukamiria** communities are located in floodplains which, during the rainy season, remain flooded for three to five months. Consequently, its agro-ecological calendar revolves around two great periods: growing and emptying. This group possess knowledge and have developed countless techniques that allow them to carry out socio-productive activities following their own calendar. Fishing, agriculture, and the gathering of wild fruits continue to be the productive activities that guarantee the livelihood of this population [47]. As for fishing, the lower part of the Marañón has great ichthyologic wealth, and thus the Kukamas enjoy a great variety of fish, among which the most frequent are zúngaros, palometa, tarpon, acarahuazú, and the carachama in their diet. Fish can realistically be said to be the primary source of protein in these communities for reasons of availability and culture.

Although hunting is not a very frequent activity, bush meat is still highly appreciated and eaten when they want to celebrate special event. The most sought-after meats are that of monkeys (such as the Colombian red howler), peccaries (such as the white-lipped peccary and the collared peccary), and big rodents (such as paca). Domestic animals are raised freely, not in pens. Birds are for self-consumption, sale, and exchange of products with other community members. Pigs are generally not eaten but are only intended for sale or trade. Regarding the consumption of fruit and vegetables, all families have one or more fields in which men and women share the responsibility of growing plants destined, for the most part, to be used as food (banana, yucca, sachapapa, sweet pepper, tomato, papaya). They also collect plants from the forest: one of these is the chonta, extracted from the trunk of palm trees and highly appreciated.

The **Urarinas** have a subsistence economy, adapted to take advantage of the natural resources of the rivers, forests, and farms. Agriculture is a complementary activity, especially for women that usually grow a multi-crop farm that serves the family as a permanent source of daily food [48]. Slashing, logging, and burning is the most common agricultural practice in all Amazonian cultures except for the Urarinas people, and that is because they are located in an area of volcanic origin, with very fertile soils. Collection is aimed at obtaining palm fruits such as aguaje and pijuayo as well as secondary forest trees such as ungurahui. Hunting is practiced individually, while fishing is practiced both individually and collectively. These activities are in direct relation to the increasing and emptying of the river. The Urarina produce poultry and agricultural products for the market. Likewise, they commercialize fine woods, leather, and palm tree fabrics—the latter in great demand on the regional market [49].

In summary, for the **Kukama** and **Urarina** populations fish is a primary food source while a secondary source comes from harvesting their gardens and the forest, supplemented by small stores located in the communities that sell products such as garlic, onion, pepper, cumin, rice, noodles, oil, sugar, and salt.

The inhabitants of the ACODECOSPAT communities are traditionally used to eating two main meals a day (74.8% of those interviewed): breakfast and lunch. After that, they only eat light foods such as rice mingado and humitas, although sometimes they do not eat anything at all, depending on whether there is something in the kitchen. Currently, some families are changing these eating habits and, when they do not go out to work all day, they eat three meals (25.2% of those interviewed).

The data that emerged from the research reveal that the main source of food for the population of the 64 communities is fish; Over ninety-three percent of those interviewed mentioned fish as the first element. Indeed, not only is it the most present food on the tables of the different communities, but it is also perceived as essential and indispensable. To the question "What are the ingredients that you try to always have in the kitchen?", 99% answered "fish". Three-quarters of those surveyed said that they eat fish every day of the week; nearly as many reported fish consumption twice a day (71.6%).

Other foods that are very present in the local diet are bananas, yucca, and rice (Figure 7 and Supplementary Materials, Section SD). Cassava, rice, bananas, papayas, corn, and cucumbers are usually grown in the fields. Respondents also reported harvesting aguaje, chonta, and medicinal plants from the forests.

Importantly, 88.5% of community members reported using river water routinely for drinking, cooking, and washing. This is not only for cultural reasons but also for a clear lack of alternatives. Only 17 out of 64 communities have a water treatment plant that cleans the water of pathogens and toxic agents (26.5%); the rest of the population uses chlorine (35.9%), boil the water (6.3%), leaves it to sediment (4.7%), or drinks it directly without doing anything (26.5%) (Figure 8 and Supplementary Materials, Section SF). Respondents also reported that during celebrations such as weddings, anniversaries, or foreign visits, they eat mainly chicken, pork, and duck.

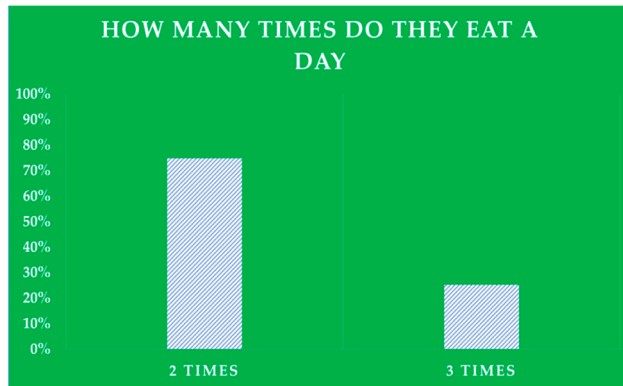
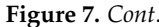
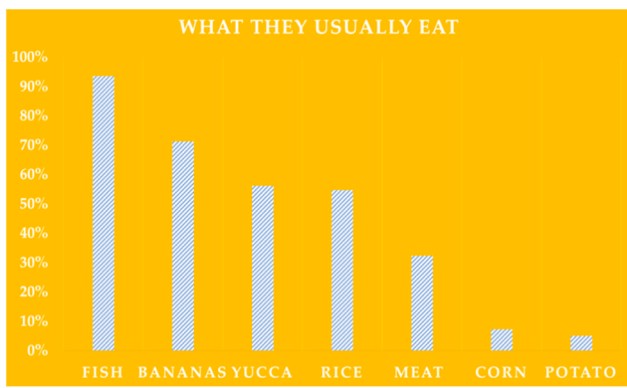

**Figure 7.** *Cont.*

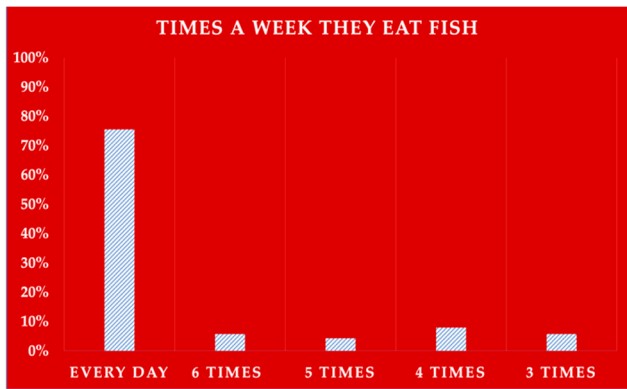
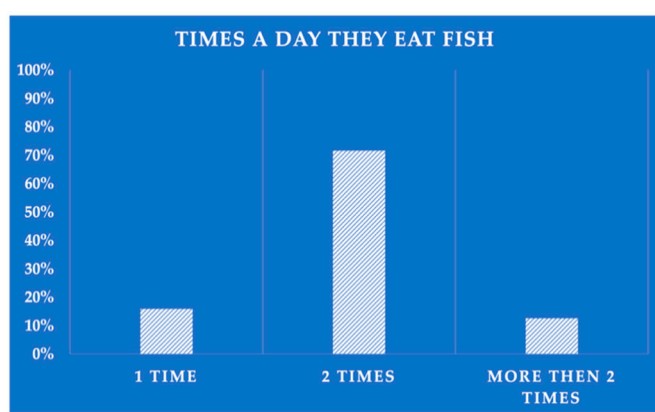

**Figure 7.** Feeding habits in the ACODECOSPAT communities (based on project's 139 surveys).

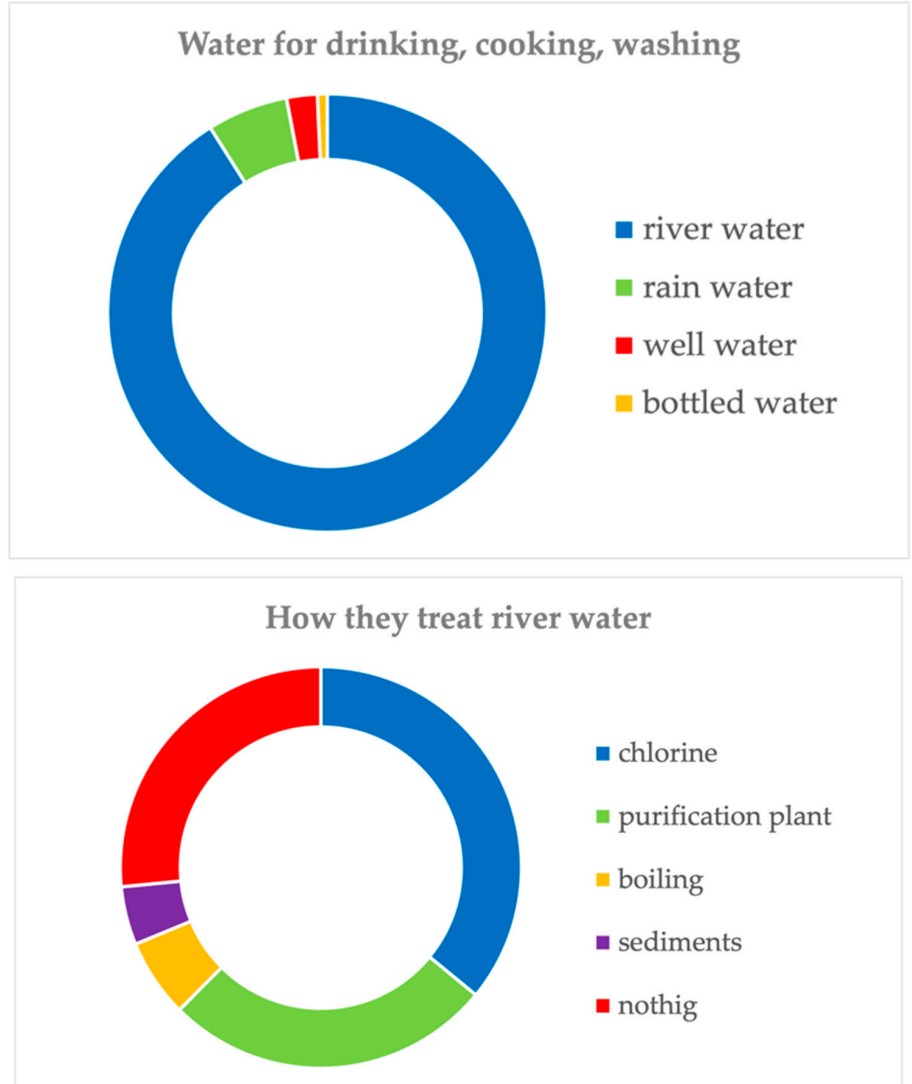

**Figure 8.** Water habits in the ACODECOSPAT communities (based on project's 139 surveys).

### 3.2. Perception of Food Insecurity

According to the data collected, the dominant type of consumption is self-consumption of fish, meat (hunted or raised), and products grown on a farm. The main products bought

in the city, in the small store that sometimes exist in the community or from street vendors, are rice, sugar, oil, salt, noodles, cookies, and soda (Supplementary Materials, Section SF, Table S12). However, few of these products represents essential goods in the diets of the various communities.

Since communities primarily consume products that they harvest themselves (fish, bush meat, and forest products), we can match the availability of food with physical access to it. With this in mind, we highlighted that the great majority of the community members have indicated a downward change in the production of both field (92%) and the fishing activity (95%). Almost 90% of the interviewees stated the change in the size and taste of the fish and 36% declare to have found black fish, with spots, with a bad taste or parasites, tumors, or much larger heads than normal. 92% complain about the size reduction of the plants in the fields and half of them state that the fruits seem burned, coming out black and shrunken.

Most interviewees blame oil pollution for this change. Indeed, 97.3% of the community members interviewed thinks that the water and fish are contaminated and 82.7% thinks that the extractive activity affects the water and the food they consume. In addition, 82.3% of the population links their diarrhea, vomiting, stomachache, headache, infections, fever, skin problems, parasitosis, and bronchitis to consuming the water and fish. One respondent commented: "even if the spills have not occurred near the community, due to the presence of the river current, the pollutants reach everyone. No one can be saved".

The data collected seem perfectly in line with the O'Callaghan-Gordo and World Health Organization studies cited in Section 1.3 that state that people's health and lives are at risk due to oil contamination and that it is possible that lead—a cause of alterations in the nervous, hematological, gastrointestinal, cardio-vascular, and renal systems—reaches Indigenous people through their diet in areas with the most serious environmental pollution rates such as the one of the Marañón River.

Based on this, it is possible to affirm that the quality of the food is strongly conditioned by the situation of contaminated water and fish, and so clearly perceived by the inhabitants. It is clear that, in a situation of self-consumption where economic access to food other than that produced is quite difficult (only 51.9% of the interviewers say they have an occasional job outside the community), the importance of the quality of food and water available is vital. Furthermore, this change in production could lead to a consequent decrease in self-consumption, causing an increase in the difficulties of access to food for those who do not have a job outside the community.

The elements reported so far, although of great interest and an excellent basis for developing possible responses to the serious problem of food insecurity of the subjects involved in the research, do not follow an internationally-recognized standard. To overcome this drawback, as explained in the previous chapter, we decided to collect more data and put the FIES method into practice. By presenting the interviewees with a set of eight questions that measure the perception of food insecurity in households, it was possible to measure the proportion of the population that faces difficulties in accessing food, at different levels of severity.

The results, summarized in Figure 9, clearly show that nearly two-thirds of the population experiences moderate food insecurity. Specifically, 65.9% of those interviewed skipped meals or saw the amount of food they consume, and portion sizes, reduced. Furthermore, 8.5% of them has felt hungry, did not or does not eat for a whole day due to lack of food produced or hunted, or lack of money to buy food.

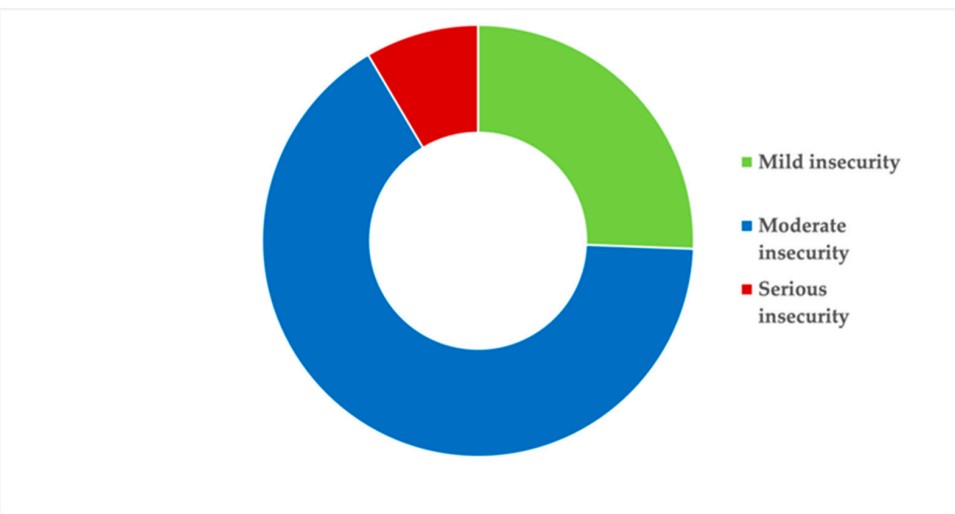

**Figure 9.** Food insecurity according to the FIES method (source: own elaboration from data collected during fieldwork).

However, the research was carried out between the months of April and October 2019, before the arrival of the SARS-CoV-2 virus in Peru. To date, we do not have reliable information that can tell us how this pandemic has affected the lifestyle and eating habits of Indigenous people.

### 3.3. Institutionality

Another aspect essential to understanding the food security of the Indigenous community of this research is institutionality. This is based on the implementation of coordinated and articulated food and nutritional security policies, in a multi-sector and intergovernmental manner that guarantees the adoption of a comprehensive vision of programs and projects. In a complex environment such as that of the Amazon rainforest, state support can be crucial for the fulfillment of the right to food.

However, from the results of the 139 interviews performed inside the ACODECOSPAT, the State does not seem to implement adequate actions that allow the population to access sustainable alternatives. Indeed, the interviewees stated that the only social program that reaches the communities of the ACODECOSPAT federation is the National School Feeding Program (Qali Warma) [50]. According to data from the Loreto Territorial Unit and from the Qali Warma National School Feeding Program [51], 60 out of the 64 communities belonging to the ACODECOSPAT federation receive care from the program. During the interviews, respondents commented that the program does not appear to be suitable for the nutritional needs of children in the communities. Specifically, they reported that sometimes the food coming from the program does not reach the community as the community's schoolteacher, who receives the food for the students, does not distribute it. In addition, children often refuse to eat distributed food because it is very different from the usual diet consumed at home.

However, projects worth mentioning are those developed by the National Program for Cooperation Fund for Social Development (FONCODES) [52] in collaboration with the same federation which, from 2016 to 2019, managed to implement actions to improve food production in 40 of the 64 communities. In addition, in October 2019 the Regional Agrarian Directorate of Loreto (DRAL) [53], in collaboration with ACODECOSPAT, registered on the inverte.pe platform, a production project with the aim of "providing technical assistance to improve food safety and nutritional quality of the diet of rural families in the districts of Nauta and Urarinas" (this project is still awaiting funding).

Even with all these measures, the food insecurity situation remains complex. Specifically, the case of the communities belonging to ACODECOSPAT would require effective

action to redeem the negative consequences generated by oil pollution, which have very dangerous repercussions on the food security of the aforementioned communities. Unfortunately, when analyzing the data collected during the fieldtrips and the interviews carried out with different officials of the national and regional institutions that deal with the issue of food security, it appears that no institution, whether national or regional, is developing adequate policies to end the vicious cycle of contamination and food insecurity.

## 4. Discussion

The concept of environmental justice (EJ) is more than a simple merger between environmental protection and social justice. This vision combined the demand for social justice and environmental sustainability with the importance Indigenous people place on the relationship between human beings and nature [54]. For Loreto Indigenous communities, culture practices, sovereignty, and immersive lives in diverse and threatened ecosystems are intertwined. One of the many environmental justice definitions states: "Environmental justice is the fair treatment and meaningful involvement of all people regardless of race, color, national origin, or income with respect to the development, implementation, and enforcement of environmental laws, regulations, and policies. This goal will be achieved when everyone enjoys the same degree of protection from environmental and health hazards and equal access to the decision-making process to have a healthy environment in which to live, learn, and work" [55].

This means that no population should bear disproportionate exposure to harmful environmental conditions, and thus environmental justice is aimed at remedying existing and imminent injustice in the distribution of environmental costs, benefits, and conditions on the grounds that all are equal and have equal rights. However, if we speak about Indigenous populations, it is important to understand the philosophical, ontological, and epistemological differences of what constitutes justice for them. For this reason, as McGregor states, "A distinct formulation of Indigenous environmental justice (IEJ) is required in order to address the challenges of the ecological crisis as well the various forms of violence and injustices experienced specifically by Indigenous peoples." [56] (p. 35). For Indigenous peoples, a just path to a sustainable future must consider all relations, not just the human but also that of non-human beings as well as of the Earth itself. This is an approach that is best expressed through Indigenous knowledge systems, legal orders, governance, and ideas of justice. As McGregor reports, "from an Indigenous point of view, environmental injustice, including the climate crisis, is therefore inevitably tied to, and symptomatic of, ongoing processes of colonialism, dispossession, capitalism, imperialism/globalization, and patriarchy" [56] (p. 36).

Following previous studies that were built on the relationship between contamination, food security, and environmental justice [57–59], my research also found that low-income communities and people of non-hegemonic ethnic groups often suffer adverse and disproportionate exposure to environmental and occupational toxins [56]. In particular, IEJ requires a combination of distributive and recognitional aspects of environmental burdens. My study's results highlighted the importance of developing a more profound understanding of the compounding effect of environmental contaminants, cultural food habits, and inadequate food access. Indeed, as stated above, Indigenous communities who live their lives following ancestral culture and lack resources have more difficulties in changing their diet and often suffer the most for the contamination of the environment they are interdependent with. By crossing the data collected on the field with information on the Kuka-Kukamiria and Urarinas people, we can conclude that we are witnessing a violation of the human right to food as a result of the high level of food insecurity connected with oil contamination. However, this is not just a problem of fulfillment of rights but also a matter of environmental injustice linked to an inequitable distribution of environmental goods and environmental risks (distributive injustice) amplified by misrecognition and general disrespect for the cultural identities, values, and knowledge systems of Indigenous groups (recognitional injustice).

In this sense, when considering that the main source of protein in these communities is fish (71.6% of those interviewed say they eat fish every day, more than once a day), the decrease in quantity and quality of this resource (which has been indicated by 95% of those surveyed) means that the current and future food security of these populations are uncertain. Furthermore, water treatments systems for river water existed for only 26.5% of the communities. Also, 65.9% of those interviewed sees a decrease in the amount of food they consume while 8.5% of them feel hungry or do not eat even for a whole day due to lack of food produced or hunted. Many interviewees blame oil contamination for decreasing and degrading their traditional foods.

Indeed, 97.3% of the community members interviewed think that their water and fish are contaminated and 82.7% thinks that extraction activity affects the water and food they consume. In addition, 82.3% of the population claims to have gotten sick with diarrhea, vomiting, stomachache, headache, infections, fever, skin problems, parasitosis, and bronchitis from consuming the contaminated water and fish. For example, the few studies carried out on the subject [5,60] found pollutants present in the soil and high lead levels in the blood of Indigenous people. But still no one is evaluating the complex socio-environmental impact that oil activity is causing in this context. Thus, the ACODECOSPAT populations are burdened with polluted water and food and bioaccumulating toxic substances likely causing health disparities. Degraded river resources reduced Indigenous peoples' capacities to access local food sources and, as Parsons reported, "can be linked to have subsequent impact on cultural values, social resilience, and Indigeneity, with the potential for cultural erosion associated with the loss of traditional relationships and interdependencies with the natural environment" [5]. In short, oil contamination and the resulting food insecurity exemplified a distributive injustice.

On the other hand, it is important to highlight that, at a national level, the issue of food security is slowly gaining ground within the political agenda. Indeed, public institutions, especially driven by international trends, are beginning to prioritize some fundamental problems related to food security that, until now, had little visibility. Similarly, at a regional level, there has been an exponential increase in projects related to food security registered between 2017 and 2019. Despite all the aforementioned, it has now been observed that the implementation of food and nutritional security policies have deficiencies in terms of public management, which often make the efforts adopted ineffective. In this context, the main problems observed are insufficient inter-institutional and inter-sector coordination and the lack of knowledge on the issue of food security, especially related to Indigenous communities. Indeed, the different actors dealing with food security often ignore others working on the issue and develop policies and projects with the same objectives. When this happens, instead of optimizing resources, efforts are duplicated without achieving concrete results. Furthermore, the lack of deep and specific technical knowledge about the scope of the problem of food insecurity is a weak factor that prevents projects from moving in the right direction [60].

By approaching the issue of food insecurity in the context of the Loreto region and by relating it to the Indigenous peoples that inhabit it, the situation seems even more worrisome. The State continues to promote and develop investment projects for the production of oil and natural gas in the area without taking into account compliance with environmental standards. This leads to continued ecosystem contamination and, consequently, a high level of food insecurity that affects Indigenous peoples. Furthermore, the few projects related to food put in place by the Peruvian state (see Section 3.3) do not seem to take into consideration the specificity of Indigenous food habits and culture. This is the sign of lack of recognition and acknowledgement that would instead be necessary to face the needs of a group of people living in such a complex context. Consistent with Parsons's [5] and Whyte's [61] work, Loreto Indigenous communities also face Recognitional Injustice.

This case study adds to a small but growing literature on Indigenous food insecurity as forms of distributive and recognition environmental injustices. Indigenous communities in Peru's Loreto region live near oil pipeline spills contaminating both land and water

resources. But the entire population of the region is at risk because the river is an open and dynamic environment. Currents carry pollutants for miles and fish, which come into contact with heavy metals, can migrate thousands of miles and reach other areas of the Amazon basin.

**Supplementary Materials:** The following supporting information can be downloaded at: https://www.mdpi.com/article/10.3390/su14126954/s1.

**Funding:** This research received no external funding.

**Institutional Review Board Statement:** Not applicable.

**Informed Consent Statement:** Not applicable.

**Data Availability Statement:** Publicly available datasets were analyzed in this study. This data can be found here: FAOSTAT at https://www.fao.org/faostat/en/#home (accessed on 12 November 2021). INEI at https://www.inei.gob.pe (accessed on 12 November 2021). Sistema de Información del Estado Nutricional—SIEN at https://www.datosabiertos.gob.pe/dataset/sistema-de-información-del-estado-nutricional-de-ni~nos-y-gestantes-perú-inscenan-instituto (accessed on 12 November 2021). World Bank Open Data at https://data.worldbank.org (accessed on 12 November 2021).

**Acknowledgments:** I would like to express my deepest thanks to Troy Abel who supported this work all the way through. Without his help, I could have not completed this work.

**Conflicts of Interest:** The author declare no conflict of interest.

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
