# Peer review of "Extractivism and Unjust Food Insecurity for Peru’s Loreto Indigenous Communities"

_sustainability, doi:10.3390/su14126954_

Round 1

Reviewer 1 Report

The subject is the entire Amazonian population is forced to face the consequences of river pollution. Some of the questions are as follows:

  1. When there is not enough food, the only indirect choice is contaminated food, starvation or chronic poisoning, which is very cruel
  2. In undeveloped countries or regions, rivers are the lifeblood of the economy. Hunting, fishermen, agricultural irrigation, and drinking water are closely related. How to solve the pollution caused by oil companies? How can poor people fight against big companies? How can environmental justice be achieved?
  3. Each ethnic group will have its own traditional culture, special lifestyle or eating style. The local aborigines follow the ancestral way of life. How can they change it? It is difficult to have a good treatment environment or method when sick?
  4. Does the Food Insecurity Experience Scale (FIES) methodological assessment work for the under-learned groups?

Author Response

Dear Reviewer 1, 

Thank you very much for your revisions. I have carefully considered the thoughtful and constructive comments and suggestions provided by the the Editor, the other two reviewers and yourself. Consequently, the manuscript has undergone thorough revisions. 
About your comments and questions, unfortunately, many are beyond the scope of my work but I did add a deeper comment In the “Material and Methods” part, as a short discussion of my assessment of FIES with Indigenous communities. I underlined the reasons why this method, designed for cross-cultural equivalence, is the best choice to conduct food security research in indigenous communities. The lack of data collected at the national institutional level regarding the food security of the Loreto population and the simplicity of the questions part of FIES make this method the most effective in a complex contest where people often don't even have access to the first level of education. Thanks to the simplicity of the questions part of FIES, it was not necessary to explain to the interviewees the complex meaning of "food security", thus avoiding directing, albeit unintentionally, answers and comments. All interviewees, regardless of their role in the community and the degree of education they possessed, were able to answer the questions without difficulty to guarantee comparable data for the research.

I very much look forward to receiving your reactions on this revised version of our manuscript.

Best regards
Caterina Rondoni

Reviewer 2 Report

DIAGNOSIS

- The content of the Paper appears in tune with the Special Issue.

- The quality standard of the Paper is not in line with the standard quality of the works published in the Review.

- The scientific approach appears, from an editorial point of view, rather elementary, both in the text and in the tables and graphics.

SUGGESTIONS

- Carefully observe the quality standard of the Works published in the Journal and in the Special Issue.

- Identify Research Collaborators, more experienced in scientific publications, capable of making that editorial improvement suitable for reaching publication. In research activities, joint work is preferred.

- Re-submit later.

Author Response

Dear Reviewer 2, 

Thank you very much for you revisions and comments. I have carefully considered the thoughtful and constructive comments and suggestions provided by the Editor, the others two reviewers and yourself. Consequently, the manuscript has undergone thorough a major revision, also following your comments.
Specifically, inspired by the Clark and Miles paper from this same special issue of the jornal, I added in the "Introduction" part a literature review utilizing the Web of Science Scholarly database that show how this paper contributes to an underexplored subtopic that overlaps food security and environmental justice, with a focus on Latin America. 
Therefore, the elementary nature of the research derives from the fact that this topic is very little explored, and I developed it thanks to field research conducted in an extreme and challenging environment. As I have repeatedly repeated in the paper, the lack of information and data on the indigenous peoples of the Peruvian Amazon rainforest is impressive. Unfortunately, few state and international investments are destined to fill these gaps. This work was created with the aim of recounting the food situation of a part of the often forgotten Peruvian population. It is impressive how the state even comes to ignore the existence of some of its citizens, who therefore fail to have access to the most basic human rights, including food. For this reason, I confirm the elementary nature of the research, underlining that it is a first step that can lead to further in-depth analysis, as demonstrated by Gonzalez's research that you have mentioned

In my opinion, the current version of the manuscript effectively attends to the issues raised and has been substantially improved due to the insightful feedback. I very much look forward to receiving your reactions on this revised version of my manuscript.

Best regards
Caterina Rondoni

Reviewer 3 Report

A well done paperwork! An interesting research results. Congratulations! My opinion is that you have to expose an abstract, in the beginning of the paperwork. 

Author Response

Dear Reviewer 3, 

Thank you very much for your revisions. I really appreciated your comment. About the abstract, you can find it at the top of the paper.

I very much look forward to receiving your reactions on this revised version of my manuscript.

Best regards
Caterina Rondoni

Round 2

Reviewer 2 Report

Dear Author, my observations did not focus on the content of the Research, which appears to be of good value, as well as on the layout and style of the presentation.

In the light of the corrections made, the work can be accepted. It is advisable, for the future, to prefer joint work (several heads are better than one). I wish the Author good progress in the Academic career.

  1. error in Affiliation “Environemtnal”
  2. line 52-53; 55-56; 61-62, it must be correct

Author Response

Dear Reviewer,

Thank you very much for your new suggestions and comments and for giving me the opportunity to resubmit a revision of my manuscript. In the future, I will take into consideration your suggestion on joint work. For the second and third comments, I corrected both.

Best regards